# The Immunogenicity of a Foot-and-Mouth Disease Virus Serotype O Vaccine in Commercial and Subsistence Cattle Herds in Zambia

**DOI:** 10.3390/vaccines11121818

**Published:** 2023-12-05

**Authors:** Frank Banda, Anna B. Ludi, Ginette Wilsden, Clare Browning, Henry L. Kangwa, Lynnfield Mooya, Masuzyo Ngoma, Geoffrey M. Muuka, Cornelius Mundia, Paul Fandamu, David J. Paton, Donald P. King, Melvyn Quan

**Affiliations:** 1Central Veterinary Research Institute, Lusaka 10101, Zambia; henrylombekangwa@gmail.com (H.L.K.); lynnemmamooya@gmail.com (L.M.); iamisinme@gmail.com (M.N.); 2Department of Veterinary Tropical Diseases, Faculty of Veterinary Science, University of Pretoria, Pretoria 0110, South Africa; melvyn.quan@up.ac.za; 3The Pirbright Institute, Pirbright GU24 0NF, UK; anna.ludi@pirbright.ac.uk (A.B.L.); ginette.wilsden@pirbright.ac.uk (G.W.); clare.browning@pirbright.ac.uk (C.B.); david.paton@pirbright.ac.uk (D.J.P.); donald.king@pirbright.ac.uk (D.P.K.); 4Department of Veterinary Services, Ministry of Fisheries and Livestock, Lusaka 10101, Zambia; muukamunkombwe@gmail.com (G.M.M.); corneliusm@hotmail.com (C.M.); pfandamu@gmail.com (P.F.)

**Keywords:** foot-and-mouth disease, immunogenicity, vaccine, field evaluation

## Abstract

The recent introduction of foot-and-mouth disease (FMD) virus serotype O (O/EA-2 topotype) in Southern Africa has changed the epidemiology of the disease and vaccine requirements of the region. Commercial and subsistence cattle herds in Zambia were vaccinated with an FMD virus serotype O Manisa vaccine according to a double- or single-dose vaccination schedule. Heterologous antibody responses induced by this vaccine against a representative O/EA-2 virus from Zambia were determined. Virus neutralisation tests (VNTs) showed double-dosed cattle had a mean reciprocal log virus neutralisation titre of 2.02 (standard error [SE] = 0.16, n = 9) for commercial herds and 1.65 (SE = 0.17, n = 5) for subsistence herds 56 days after the first vaccination (dpv). Significantly lower mean titres were observed for single-dosed commercial herds (0.90, SE = 0.08, n = 9) and subsistence herds (1.15, SE = 0.18, n = 3) 56 dpv. A comparison of these results and those generated by solid-phase competitive ELISA (SPCE) tests showed a statistically significant positive correlation by Cohen’s kappa coefficient. Therefore, SPCE might be used in assessing the immunogenicity of vaccines in place of VNT. Furthermore, for this vaccine and field strain, a vaccination regime employing a two-dose primary course and revaccination after 4–6 months is likely to be appropriate.

## 1. Introduction

Foot-and-mouth disease (FMD) is one of the most economically important infectious diseases of livestock in the Southern African Development Community (SADC) due to its impact on livestock productivity and resulting international trade restrictions on live animals and livestock products. The causative agent, FMD virus (FMDV), is contagious and antigenically diverse, with six currently circulating serotypes that do not cross-protect. This role of virus diversity in the epidemiological dynamics of FMD in Africa has been described [1].

There has been an escalation in FMD outbreaks in the past decade in southern Africa [2] with Zambia and Namibia reporting novel serotype O incursions due to the O/EA-2 topotype [3]. Before 2018, FMD was confined to three FMD high-risk areas of Zambia [4,5,6], but, in 2018, an FMDV serotype O outbreak was reported in the Chisamba District of Central Zambia [3]. This disease spread to all the provinces in Zambia except the Luapula Province. A subsequent spillover of this outbreak to the Zambezi Region of Namibia during June–July 2021 was also recorded [3]. Further spread of this lineage into Malawi and northern Mozambique was reported in 2022 [7,8].

Vaccination campaigns together with stringent control measures have been used to eradicate FMD from Europe and most of South America [9,10]; in endemic regions, vaccination, especially with good quality FMD vaccines, is an important means of FMD control and can help prevent losses in stock production and reduce the overall incidence of the disease [11]. Post-vaccination monitoring (PVM) guidelines have been published to advise countries on principles and suggested procedures for monitoring various aspects of FMD vaccination [12]. In Southern Africa, FMD vaccination has targeted livestock mainly for prophylaxis and to prevent the spread of the disease using trivalent vaccines containing Southern African Territories serotypes (SAT-1, SAT-2, and SAT-3) antigens [13]. The introduction of serotype O into the region means that new vaccines are needed to control outbreaks due to this serotype. As there are no empirical data available to show the appropriateness of serotype O vaccines in Zambia, this study was conducted using the principles of the PVM guidelines to assess the immunogenicity of an imported serotype O vaccine in Zambia.

The specific objectives of this study were to evaluate FMDV-specific antibody responses after a single dose and a second booster dose in vaccinated cattle and to compare results obtained by the virus neutralisation test (VNT) and a commercial solid-phase competitive ELISA kit (SPCE).

## 2. Materials and Methods

### 2.1. Study Design

Field studies were conducted in commercial cattle (Study A) and in subsistence cattle (Study B) to assess the immunogenicity of an imported serotype O vaccine using either a one- or two-dose primary vaccination protocol. Study A animals were Friesian cattle from a commercial dairy farm in the Momboshi area of the Chisamba District, Zambia, whilst Study B animals were mixed breeds in herds from different subsistence farmers in multiple sublocations in the Rufunsa District, Zambia. There was no previous history of any FMD vaccination in either of these areas, and there was no history of FMD outbreaks in the previous two years. Selected animals were six to twelve months old at vaccination and included both males and females. All animals were individually identified by ear tags to ensure accurate follow-up. Consent was obtained from the farmers to use their animals in the study and owners were advised not to move their animals out of the farms during the study. This project was approved under research ethics committee number REC 061-19 from the University of Pretoria, South Africa.

The protocol followed the Food and Agriculture Organization and World Organisation for Animal Health (FAO-WOAH) PVM guidelines Section 3.3 [12]. In Study A, 69 animals were recruited, including ten unvaccinated controls to act as sentinels for FMDV exposure, whilst in Study B, 55 animals were recruited, including five unvaccinated controls. Sera from 32 cattle were randomly selected and tested for antibodies to FMDV for each of the five time points (0–168 days after the first vaccination; dpv).

Of the vaccinated animals in both studies, ten were given a second dose 28 days post-vaccination (dpv). A second dose is recommended by the vaccine manufacturer and previous research on immunologically naïve animals (i.e., with no previous FMDV exposure or vaccination) [14]. Blood samples were collected in plain Vacutainer^®^ tubes (Becton Dickinson, Franklin Lakes, NJ, USA) at first vaccination (0 dpv), 28 dpv (time of second vaccination), 56 dpv, 112 dpv, and 168 dpv.

Not all samples were tested for the presence of FMDV antibodies because the budget for this study was limited. From Chisamba, samples from nine single-dose and nine double-dose cattle were tested as well as samples from four unvaccinated controls. From Rufunsa, samples from three single-dose, five double-dose, and two unvaccinated control cattle were tested.

### 2.2. Vaccine

The animals were vaccinated with a monovalent FMDV serotype O Manisa vaccine of at least three PD_50,_ containing purified inactivated FMDV antigen and aluminium hydroxide with saponin as an adjuvant, obtained from the Botswana Vaccine Institute. This aqueous adjuvanted vaccine is indicated for use in cattle, buffalo, sheep, and goats. Each dose (1 mL) was given according to the recommendations from the vaccine manufacturer. In a primary course of vaccination, two injections three to four weeks apart are recommended, whilst a booster dose is advisable every four to six months depending on the risk and following local practice.

### 2.3. Serological Testing

All samples were shipped on ice to the Central Veterinary Research Institute (CVRI), Zambia, and, on arrival, sera were separated and then stored at −20 °C until testing. All samples were tested with an FMDV non-structural protein (NSP) ELISA kit (IDEXX, Westbrook, MN, USA) according to the manufacturer’s instructions at the CVRI to evaluate if natural exposure to FMDV had occurred during the study period. Antibodies to viral NSPs are considered a reliable indicator of evidence of previous or current viral replication in the host, irrespective of vaccination status [15,16,17]. Test results for the NSP ELISA are expressed as a percentage positivity relative to the strong positive control [(optical density of test or control wells/optical density of strong positive control) × 100] [18]. Values < 20% are considered negative, values ≥ 20% and <30% are considered suspect, and values ≥ 30% are considered positive. 

Samples were also tested using a solid phase competitive ELISA (SPCE) or antibodies specific to FMDV serotype O structural proteins (IZSLER Biotechnology Laboratory, Brescia, Italy). This test detects antibodies elicited by vaccination and natural infection. A percentage inhibition for this test is calculated for each well (100 − [optical density of each test or control value/mean optical density of the 0% competition] × 100%), representing the competition between the test sera and a specific murine monoclonal antibody for the FMDV antigen on the ELISA plate. Using the semi-quantitative method applied in this test, sera are considered positive when there is inhibition ≥ 70% and negative when inhibition is <70% at 1:10 dilution. A second dilution (1:30) of strongly positive sera samples (≥80% inhibition at 1:10 dilution) indicates the level of antibodies with strongly positive sera showing ≥80% inhibition at both 1:10 and 1:30 dilutions.

Samples were shipped on dry ice to the FAO World Reference Laboratory for FMD (WRLFMD), Pirbright, UK, for testing by VNT, as described previously [15,19]. A representative Zambian field isolate from the O/EA-2 topotype (O/ZAM/7/2021) was used in the VNT to derive field strain-specific (heterologous) titres. Earlier comparison of the neutralisation of O Manisa and O/ZAM/7/2021 viruses by O Manisa post-vaccination sera had shown an antigenic similarity score (r_1_ matching value) of 0.47, where 1.0 is a perfect match and values greater than or equal to 0.3 are considered indicative of an acceptable match for field vaccination [20]. The post-vaccination VNT titres to heterologous viruses associated with post-vaccination cross-protection have been evaluated. Using VNT, an indicator of likely heterologous cross-protection is considered to be a log_10_ reciprocal titre of 1.5 (1:32) after a single-dose vaccination with serum collected 21 days later [21,22].

### 2.4. Data Analysis

Microsoft Office Excel 2016 (Microsoft Corporation, Redmond, WA, USA) and SPSS Statistics v29 (IBM, Armonk, NY, USA) were used to analyse the data.

## 3. Results

During the study, one animal died in Study A, and five cattle died or went missing in Study B. The animal that died in Study A belonged to the unvaccinated control group. From Study B, deaths were observed at different time intervals during the study, with one animal dying from each group, i.e., unvaccinated controls, single dosed, and double dosed. Their deaths were attributed to East Coast fever, a tick-borne disease which is prevalent in some areas due to poor dipping practices, especially by farmers from Study B. This was supported by pathological signs which included fever, enlarged lymph nodes, anorexia, laboured breathing, corneal opacity, nasal discharge, diarrhoea, and anaemia. The two missing animals could have been sold by the farmers, as in this sector, animals serve as a source of income, food, and social security [23].

Although three cattle tested weakly positive for NSP antibodies at single time points in the study all animals eventually tested negative for NSP antibodies, and there was no evidence of clinical FMD in the study areas.

There were differences in the VNT titres between cattle receiving one and two doses of vaccine (Appendix A and Figure 1). The mean logs of the reciprocal VNT titres 56 dpv were 1.15 (standard error of the mean [SE] = 0.18, n = 3) and 1.65 (SE = 0.17, n = 5) for subsistence cattle receiving 1 and 2 doses of vaccine, respectively, and 0.90 (SE = 0.08, n = 9) and 2.02 (SE = 0.16, n = 9) for commercial cattle receiving 1 and 2 doses of vaccine, respectively. The mean titre obtained from the samples of unvaccinated animals was 1.01 log_10_ (range 0.6–1.2). Only one subsistence animal (out of three) and no commercial animals (out of nine) that were vaccinated once developed VNT titres by 56 dpv above the threshold value of 1.5 to be considered protective, but three subsistence animals (out of five) and seven commercial animals (out of nine) that were vaccinated twice developed VNT titres by 56 dpv to be considered protective.

The pattern of structural proteins binding antibody responses largely mirrored those for neutralising antibodies. Peak responses in double-dose cattle were detected 56 days after the first vaccination. The mean SPCE percentage inhibitions 56 dpv were 25.7 (SE = 12.6, n = 3) and 86.6 (SE = 8.8, n = 5) for cattle from Rufunsa receiving 1 and 2 doses of vaccine, respectively, and 14.1 (SE = 5.6, n = 9) and 60.6 (SE = 8.6, n = 9) for cattle from Chisamba receiving 1 and 2 doses of vaccine, respectively. None of the animals from Rufunsa (out of three) and only one animal from Chisamba (out of nine) that were vaccinated once developed antibodies by 56 dpv above the threshold value of 70% inhibition to be considered positive, but four animals from Rufunsa (out of five) and all animals from Chisamba (out of nine) that were vaccinated twice developed antibodies by 56 dpv to be considered positive (Appendix A, Figure 1). A one-sided independent samples *t*-test where equal variance was assumed showed a significant difference in antibodies 56 dpv between cattle given one and two doses of vaccine (Rufunsa: *p* = 0.003; Chisamba: *p* < 0.001).

A Spearman’s rank-order correlation determined the relationship between the log of the reciprocal of the virus neutralisation titres and the SPCE % inhibition results (Figure 2). There was a positive correlation between the VNT and SPCE results, which was statistically significant (ρ = 0.598 for 1:10 dilution, and ρ = 0.562 for 1:30 dilution of SPCE % inhibition results, *p* < 0.001 for both dilutions). 

Statistical analysis by Cohen’s kappa coefficient showed a moderate agreement [24,25] between the two assays (κ = 0.516, 95% CI, 0.365 to 0.667, *p* < 0.001) using positive titre thresholds of ≥1:32 for the VNT and ≥70% inhibition for the SPCE.

## 4. Discussion

In this study, the immunogenicity of an imported FMD vaccine was evaluated to determine its suitability for use in Zambia and other Southern African countries where serotype O has been described as becoming important [26]. The evaluated vaccine is already widely used in sub-Saharan Africa and was an obvious candidate to control the FMDV serotype O outbreaks reported in the Southern African region. FMDV-specific antibody responses were measured after one- or two-dose primary courses of the vaccine. The study also compared results obtained by the VNT and a commercial SPCE kit to evaluate the use of simple-to-use tests to support PVM studies at a population level in settings where high-containment facilities are not available.

The performance of an FMD vaccine can be assessed in vaccination-challenge studies according to the protocols defined in the WOAH Terrestrial Manual [15]. However, due to several reasons including ethical considerations and costs, in vitro studies are recommended [15]. The relationship between serology and protection in FMD-vaccinated cattle has been previously studied by correlating the antibody titres at the point of challenge with the outcome, i.e., protected or not protected [21,27,28]. A correlation between neutralising antibody titre and protection against homologous challenge is well established [29,30]. The testing and analysis of day-of-challenge sera from vaccination- and challenge-cross-protection studies have long established an association between in vitro neutralising antibody titres to the challenge viruses and in vivo clinical cross-protection [27]. Cross-strain protection between different vaccines and challenge viruses also have similar relationships, although much less data are available from these studies [21,27]. A recommended simple approach to assess cross-protection is to measure the amount of antibodies that vaccinated animals have against the field virus of concern [27]. It is against this background that this study examined day-of-challenge antibody titres to a heterologous challenge virus.

Apart from the three weakly positive samples observed at one time point, neither seroconversion to FMDV NSP nor clinical signs were observed throughout this study, suggesting the FMDV-specific antibody titres detected were vaccine-induced. The observed mean reciprocal log virus neutralisation titre of 2.02 for commercial cattle from Chisamba and 1.65 for subsistence cattle from Rufunsa in second-dosed cattle (Appendix A and Figure 1) imply that the aqueous adjuvanted vaccine used in this study produced neutralisation titres likely to offer protection against the target lineages. However, the observed mean titres from single-dosed cattle from Chisamba (0.90) and Rufunsa (1.15) 56 days after the first vaccination suggest that the vaccine given in this way was not able to reach the recommended protective titres as described [21,22]. This emphasises the value of the second vaccine dose in this context, but its importance is likely to vary according to the potency of the vaccine and its antigenic match to the strain against which protection is required. After the primary course of vaccination, the manufacturer of this vaccine recommends a booster at 4–6 months, which also seems in line with the results obtained here on the duration of the antibody responses found via VNT and ELISA. Apart from these general trends, care must be taken not to overinterpret the findings, as predicting protection from antibody titres cannot be carried out with precision due to several uncontrollable variables [21]. Other studies [30] have also shown that antibodies alone are not responsible for protection but are a correlate of the immune response in the animal.

Previous studies have correlated potency test outcomes with pre-challenge antibody titres measured using the VNT and LPBE [29,31]. More recently, commercially available SPCEs have become alternatives to the LPBE for SP serology and use for routine PVM at the population level [12,32]. A limitation of SPCE in having antigens and blocking antibodies which are fixed and usually of unknown antigenic relevance (beyond serotype) has, however, been discussed [32], and this together with antigen instability [33] may account for differences in results compared to VNT. Nevertheless, studies have shown the potential of SPCE in the determination of immunogenicity [34]. The moderate correlation in results between the two test methods (i.e., VNT and SPCE) in this study suggests that the SPCE could be used in assessing the immunogenicity of vaccines where high-level biocontainment facilities are not available to conduct VNT, although the VNT is likely to be superior at predicting cross-protection to specific field strains according to their antigenic match.

## 5. Conclusions

This study confirms the value of PVM testing and shows that for this vaccine and field strain, a vaccination regime employing a two-dose primary course and revaccination after 4–6 months is likely to be appropriate, as recommended by the vaccine manufacturer. Countries, especially sub-Saharan countries with a diversity of FMDV strains and inconsistent vaccine quality, are encouraged to implement PVM studies that can account for variations in vaccine potency and antigenic match.

## Figures and Tables

**Figure 1 vaccines-11-01818-f001:**
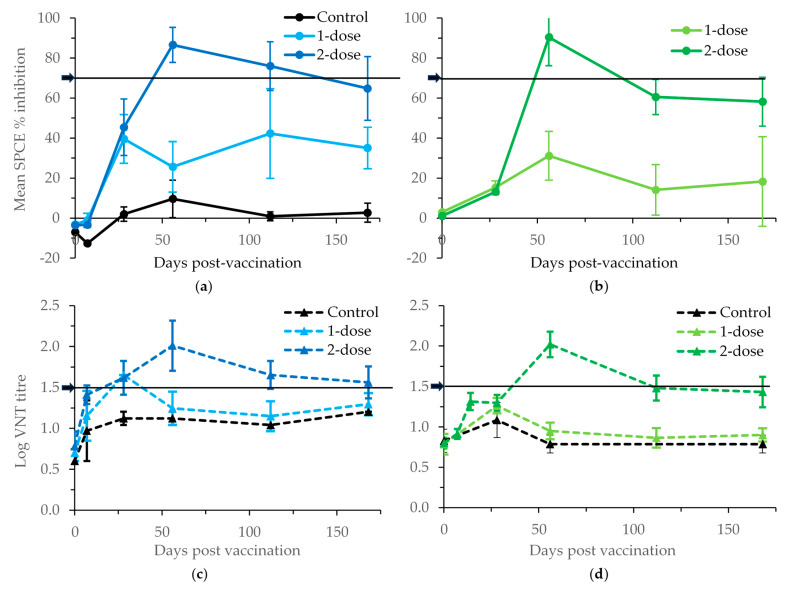
The mean solid-phase competition ELISA (SPCE) % inhibition at 1:10 serum dilution (**a**,**b**) and virus neutralisation test (VNT) titres (**c**,**d**) in subsistence (**a**,**c**) and commercial (**b**,**d**) cattle herds vaccinated against FMDV serotype O at day 0 only (1-dose) or again 28 days after the first dose (2-dose). Control animals were not vaccinated. Values greater than the threshold values indicated on the *y*-axis by arrows are considered protective.

**Figure 2 vaccines-11-01818-f002:**
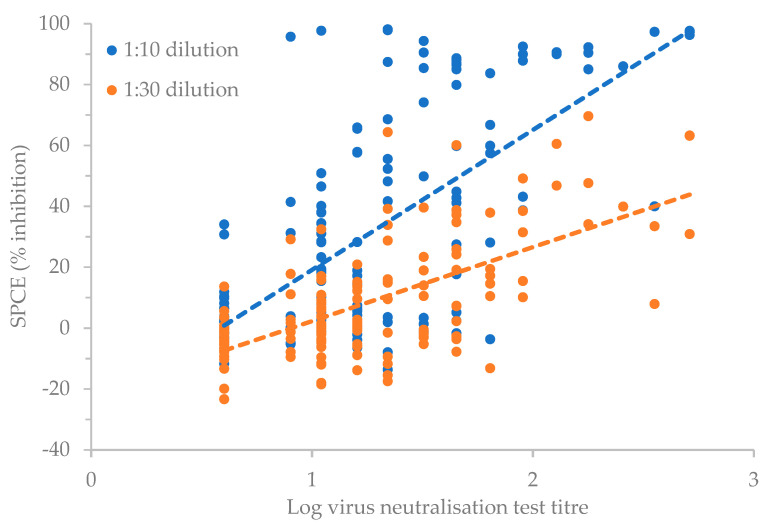
Positive correlation between log_10_ virus neutralisation titres and 1:10 (blue, R^2^ = 0.446) and 1:30 (orange, R^2^ = 0.426) solid-phase competition ELISA (SPCE) % inhibition in cattle vaccinated against FMDV serotype O.

## Data Availability

All datasets produced and analysed for this study are available from the corresponding author upon reasonable request.

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
