# Peer review of "The Immunogenicity of a Foot-and-Mouth Disease Virus Serotype O Vaccine in Commercial and Subsistence Cattle Herds in Zambia"

_vaccines, 2023, doi:10.3390/vaccines11121818_

Round 1

Reviewer 1 Report

Comments and Suggestions for Authors

Response to authors

This manuscript describes a finding that is relevant for vaccination in the field in Africa, where this disease is endemic. The single vaccination strategy is often used due to costs of the vaccination. It is important to stress the need for doing prime-boost (double) vaccinations using these locally produced vaccines in a natural setting. As such this work deserves publishing.

I find the legend to Table S1 having a duplication. It explains the colouring of the VNT at the bottom, using log10 VNT titres, similar to the values in the Table itself. However, in the text at top it says

“The SPCE results are semi-quantitative with a darker shade of green indicating stronger positivity and negative results indicated in grey. The positive VNT results are indicated in light green for partially protective titres (> 11; < 32), green for a protective titre (32) and dark green for strongly protective titres (> 32); negative results are indicated in light grey.”

I first dislike that titres of VNT are here mentioned as plain titres (not log10), which makes it difficult to compare with values in the Table. Furthermore, this text is a duplication of the legend below, especially with respect to VNT. I therefore recommend to replace this text with:

“The SPCE and VNT results are semi-quantitative represented with a darker shade of green indicating stronger positivity and negative results indicated in grey, as explained in the legend below the Table.”

Figure 1 and Figure 2: I dislike the lines of the axes and tick marks being grey. Make them black. This is typically something Excel always does wrong….

Figure 1: I prefer that a line is added in Figure 1 that indicates the cutoff/threshold for protection, rather than the thick arrow. This line makes it easier to see if values at the right of the figure are above or below this threshold. This is relevant, since the values are close to the threshold.

Lines 157-160 says:

“Only one subsistence animal (out of three) and no commercial animals (out of nine) that were vaccinated once developed protective titres by 56 dpv above the threshold value of 1.5 to be considered positive but three subsistence animals (out of five) and seven commercial animals (out of nine) that were vaccinated twice developed protective titres by 56 dpv to be considered positive.”

I find this sentence a bit confusingly written with respect to ‘protective’ and ‘positive’. I suggest to replace ‘protective titres’ by ‘VNT titres’, which is more objective. I also suggest to replace ‘positive’ by ‘protective’.

Line 163-164: I dislike the start of the sentence with ‘However’ and the use of the word ‘but’ later. Perhaps it can be rephrased.

References 7, 14 and 19 are websites. Please indicate the day accessed also for ref 19. Furthermore, please check reference 14 for description of the site. Please remove the year indications…… I checked the instructions to authors for this website referencing…it says this:

9. Title of Site. Available online: URL (accessed on Day Month Year).
Unlike published works, websites may change over time or disappear, so we encourage you create an archive of the cited website using a service such as 
WebCite. Archived websites should be cited using the link provided as follows:
10. Title of Site. URL (archived on Day Month Year).

Please follow these instructions.

In line 136-137 there is also a reference to a website that is simply given as URL rather than a reference visible in the references list. Please change this also into a proper reference to a website.

I find in general that a lot of references are given to websites. I also dislike reference 2, which is to a proceedings of a conference, which I find difficult to retrieve from the internet. I suggest to simply delete reference 2, since it is not essential for this manuscript.

Reference 12 is also not a true journal, but a link to the FAO website. It has two ISBN numbers. That is confusing. I propose to provide a proper reference for this approach. A lot of publications are available for post vaccination serology by ELISA. I here provide a few:

1.    It has been suggested that measuring IgG1 or IgG2 in cattle improves correlation with protection against FMDV by Brito et al. Brito BP, Perez AM, Capozzo AV. 2014. Accuracy of traditional and novel serology tests for predicting cross-protection in foot-and-mouth disease vaccinated cattle. Vaccine 32:433–436. http://dx.doi.org/10.1016 /j.vaccine.2013.12.007.

2.    Others used liquid phase blocking ELISAs to assess serology. They also found better correlations with VNT. See e.g. Van Maanen C, Terpstra C. 1989. Comparison ofa liquid-phase blocking sandwich ELISA and a serum neutralization test to evaluate immunity in potency tests of foot-and-mouth disease vaccines. J. Immunol. Methods 124:111–119. http://dx.doi.org/10.1016/0022-1759(89)90192-0

3.    Massive use of a SPCE using Dutch sera: Chenard G, Miedema K, Moonen P, Schrijver RS, Dekker A. A solid-phase blocking ELISA for detection of type O foot-and-mouth disease virus antibodies suitable for mass serology. J Virol Methods. (2003) 107:89-98.

4.    Mansilla et al compared various LPBE and SPCE ELISAs to VNT and showed that 146S integrity is important as well as found correlation with IgG1, again. Mansilla FC, Turco CS, Miraglia MC, Bessone FA, Franco R, Perez-Filgueira M, et al. The role of viral particle integrity in the serological assessment of foot-and-mouth disease virus vaccine-induced immunity in swine. PLoS ONE. (2020) 15:e0232782. doi: 10.1371/journal.pone.0232782.

5.    A review was written about the topic: Paton DJ, Reeve R, Capozzo AV, Ludi A. Estimating the protection afforded by foot-and-mouth disease vaccines in the laboratory. Vaccine. (2019) 37:5515-24. doi: 10.1016/j.vaccine.2019.07.102. I suggest to give this reference at least at the position where reference 12 is now, replacing reference 12, both in Introduction and discussion.

Line 191 “was run to determine” is a bit of labtalk. Please rephrase.

I find it a bit confusing that the correlation coefficient (R^2) is given in Figure 2 while line 189 says “ρ = 0.598 for 1:10 dilution, and ρ = 0.562 for 1:30 dilution”. I would prefer that the text matches the Figure and I have a clear preference for R^2 since I understand this value and not the ρ. Sorry.

I find the correlation between VNT and SPCE quite low. Others found a much better correlation, with R^2 values of 0.8 to 0.9. I think the discussion of this topic at the end of the discussion is very superficial. I would prefer the authors discuss this correlation between SPCE and VNT in more detail. Since so many scientists from TPI are present on this paper I am a bit surprised by this. Is this poor correlation between VNT and SPCE due to different antigens (FMDV strains) used in VNT and SPCE? Is it due to presence of 12S in SPCE? Which epitope does the mAb used in SPCE bind to? Is it the GH-loop  that is also highly immunogenic and important for VNT? Is this correlation of 0.42 to 0.44 really sufficient for using it for PVM?

Please use the above questioning and references given to more thoroughly discuss the correlation between VNT and SPCE.

Reviewer 2 Report

Comments and Suggestions for Authors

Foot-and-mouth disease virus (FMDV) is a single-stranded positive-sense RNA virus that causes foot-and-mouth disease (FMD) in domestic and wild cloven-hoofed animals worldwide, including swine, sheep, goats, cattle, camelids, and deer. Frank Banda et al described the immunogenicity of a foot-and-mouth disease virus serotype O vaccine in commercial and subsistence cattle herds in Zambia. This study confirms the value of PVM testing and shows that for this vaccine and field strain, a vaccination regime employing a two-dose primary course and revaccination after 4-6 months is likely to be appropriate. Overall, this article is interesting and the results may lay the foundation for FMD control. I made some suggestions that might improve the article.

1. In the abstract section, the author writes without logic and lacks a conclusion.

2. In the introduction section, the author should provide more information.

3. In the results section, the author should divide this part into several sections and provide more figures or tables.

4. In the References section, there is a lack of recent literature.

Reviewer 3 Report

Comments and Suggestions for Authors

In this manuscript titled “The immunogenicity of a foot-and-mouth disease virus serotype O vaccine in commercial and subsistence cattle herds in Zambia”, the authors tested the antibody response against serotype O FMD after vaccination via two assays, the virus neutralisation test (VNT) and a commercial solid-phase competitive ELISA kit (SPCE). The authors found that two-dose vacation, which is manufacturer’s preferred vaccination schedule, had a significantly higher response that one-dose vacation and control. Overall, this is an interesting small-scale study. However, due to the limited number of samples tested, I am not very confident in the conclusion of this study. Therefore, I would recommend a major revision of this manuscript.

Major issues:

1. In “2.1. Study Design” section, the authors mentioned 69 animals were enrolled in study A and 55 animals were enrolled in study B, however, only 32 animals were selected for test, which is only about 25% of total animals enrolled. The authors stated that “not all samples were tested for the presence of FMDV antibodies because the budget for this study was limited”. I would highly recommend testing the rest of animals enrolled in the two studies at a single time point after vaccination and with one assay the authors preferred. This would significantly reduce the authors’ budget of testing all the animals. At this stage, due to the number of animals tested per group is too small, I am not sure the conclusion can hold in real world.

2. Line 146-147, “During the study, one animal died in Study A and five cattle died or went missing in Study B. Their deaths were attributed to tick-borne diseases”. More details are needed for the animals died early. Which treatment groups they belonged to? When did they die after vaccination? What is the evidence supporting that they were died from tick-borne diseases? The authors could add the details into the method section or put them into a supplemental table.

3. Line 161-163, “A one-sided independent samples T-test where equal variance was assumed showed a significant difference in titres at 56 dpv between commercial cattle given one- and two-doses of vaccine (p < 0.001).” T-test is improper here, because additional control group and time points are involved in this dataset. Two-way ANOVA is needed for the statistical analysis in Figure 1.

Round 2

Reviewer 3 Report

Comments and Suggestions for Authors

All concerns resolved